# Energy-Saving Adaptive Routing for High-Speed Railway Monitoring Network Based on Improved Q Learning

**DOI:** 10.3390/s23177393

**Published:** 2023-08-24

**Authors:** Wei Fu, Qin Peng, Canwei Hu

**Affiliations:** Key Laboratory of Industrial Internet of Things and Network Control, Ministry of Education, Chongqing University of Posts and Telecommunications, Chongqing 400065, China; pengqin1998@gmail.com (Q.P.); canweihu9527@gmail.com (C.H.)

**Keywords:** high-speed rail, wireless monitoring system, routing, Q-learning, lifetime, latency

## Abstract

In high-speed railway operational monitoring network systems targeting railway infrastructure as its monitoring objective, there is a wide variety of sensor types with diverse operational requirements. These systems have varying demands on data transmission latency and network lifespan. Most of the previous research focuses only on prolonging network lifetime or reducing data transmission delays when designing or optimizing routing protocols, without co-designing the two. In addition, due to the harsh operating environment of high-speed railways, when the network changes dynamically, the traditional routing algorithm generates unnecessary redesigns and leads to high overhead. Based on the actual needs of high-speed railway operation environment monitoring, this paper proposes a novel Double Q-values adaptive model combined with the existing reinforcement learning method, which considers the energy balance of the network and real-time data transmission, and constructs energy saving and delay. The two-dimensional reward avoids the extra overhead of maintaining a global routing table while capturing network dynamics. In addition, the adaptive weight coefficient is used to ensure the adaptability of the model to each business of the high-speed railway operation environment monitoring system. Finally, simulations and performance evaluations are carried out and compared with previous studies. The results show that the proposed routing algorithm extends the network lifecycle by 33% compared to the comparison algorithm and achieves good real-time data performance. It also saves energy and has fewer delays than the other three routing protocols in different situations.

## 1. Introduction

With the rapid development of high-speed railway (HSR) networks, higher requirements have been placed on the safety and stability of high-speed trains, and the train operating environment is directly related to the safety of high-speed trains in transit. Ways to quickly find and solve basic faults is the focus of HSR systems around the world, and high-speed train operation environment monitoring is an important guarantee for railway safety operations [1]. The safety of high-speed railway systems can be significantly improved through real-time monitoring and inspection. The high-speed railway operation environment monitoring system includes the railway infrastructure monitoring system, the high-speed railway natural disaster and foreign body intrusion monitoring system, the EMU on-board dynamic monitoring system, and the EMU operation and maintenance management system. At present, the wired communication network adopted by these monitoring systems has been verified to be stable and reliable [2,3]. However, the high-speed railway operating environment is complex, and the high cost of wired network is not conducive to large-scale deployment of the global high-speed railway monitoring system, and the complex terrain is inconvenient for wired network maintenance [4,5,6,7].

The development and maturity of wireless sensor network technology provides more efficient and reliable, low-cost, easy-to-implement and maintain, high-tech means for the field of high-speed train operation environment monitoring. The use of wireless systems to monitor the operating environment of high-speed railways can realize large-scale deployment along the railway, ensure the breadth and accuracy of data collection, and monitor the condition of slopes, tunnels, roadbeds, bridges, and other facilities, which can effectively meet the needs of high-speed trains. The need for comprehensive monitoring of the operating environment reduces the cost of train operating environment testing and is suitable for large-scale deployment and long-term online monitoring in key areas and remote areas along the high-speed railway. However, the energy resources of wireless devices are strictly limited. The greatest challenge in wireless sensor networks is determining a way to conserve energy, and the amount of time that the network lasts is a good measure of teh quality of its performance [8,9].

In order to ensure the real-time performance of high-speed railway operating environment monitoring information and the energy utilization efficiency of the network system, a protocol that adapts to the characteristics of the high-speed railway operating environment monitoring network is needed to efficiently utilize the limited energy resources of the network system and provide more long-term energy efficiency [10]. The increase in communication distance increases the energy consumption of some nodes and shortens the network life. Each protocol chooses to increase the number of forwarding hops to achieve a balance of energy consumption, and the overall energy consumption is evenly distributed to more nodes to avoid premature death of some nodes. But the increase in hop count inevitably increases the delay of information transmission, which is bad news for the high-speed railway monitoring network [11,12]. In fact, it is difficult to guarantee the real-time performance of the network and its maximum lifespan at the same time. Extending the lifespan and increasing the number of hops results in a large delay and affects the service quality of the monitoring network. To this end, in the intelligent high-speed railway monitoring system. The information in the network can be divided into several parts: receiving, transmitting, processing, state evaluation and prediction, and control decision-making. Generally speaking, the transmission of data along the shortest path can minimize the energy consumption of the network, but this approach introduces the problem of unbalanced energy consumption. The energy consumption of the sensor device closer to the sink node is faster, which is the so-called energy hole phenomenon [13,14,15].

This phenomenon destroys the balance of energy consumption among nodes, affects the life of the network, and also has a negative impact on the real-time performance of data transmission, hindering the normal service of the network system. Therefore, in addition to minimizing the network energy consumption, the energy consumption balance among network nodes should also be considered when designing the routing algorithm to extend the life of the network system. Therefore, when designing a routing protocol for a high-speed railway operation environment monitoring network, it is necessary to minimize and balance network energy consumption while reducing data transmission delay so as to improve real-time performance and network energy utilization, and prolong network service life. Therefore, the goal of this paper is to look into ways in which a high-speed railway operation environment monitoring network can use an adaptive routing method to meet the needs of different services [16,17].

Table 1 introduces some typical existing studies and evaluates them in terms of optimization goals, adoption methods, network structures, and advantages and disadvantages. Most of the research is aimed at improving the lifetime of network systems, and the strategies to maximize network lifetime can be divided into deployment optimization, data processing, and protocol design. Among them, routing protocol design optimization is a more effective and widely used strategy, and protocol design can be divided into single-hop, multi-hop, and clustering methods. However, for a typical linear network such as a high-speed railway operating environment monitoring network, it is obviously more appropriate to use a multi-hop routing protocol. In some studies, the network lifetime is extended by minimizing the total network energy consumption (MTECR). However, its shortcomings are also obvious. As mentioned above, the problem of unbalanced network energy consumption while minimizing energy consumption leads to network energy holes, resulting in the premature death of some nodes, which affects the service life of the network. In addition, some other research focuses on balancing the energy consumption of nodes in the network, such as MVECR and AUMRP, and proves that their balancing scheme based on the residual energy of nodes is more effective for prolonging the network lifetime.

To the best of our knowledge, there are few studies on the co-optimization of lifetime and delay for high-speed railway operational environmental monitoring networks. Traditional routing algorithms cannot ensure stable transmission services in dynamic environments because they cannot cope with network dynamics and voids. The maintaining of global routing information incurs great overhead, and the complex structure also reduces the efficiency of the current high-speed railway linear monitoring network. These issues are addressed in this article.

## 2. Motiviation

Power supply and grid maintenance for high-speed rail operating lines are very difficult. With the development and maturity of wireless sensor network technology, high-speed train operation provides more efficient, reliable, and low-cost solutions in the field of environmental monitoring and detection, and then implements high-tech manual maintenance. Experts have conducted a lot of in-depth research on the coordination of the routes of the plane network, but less on the coordination of the high-speed railway network. If there is a general problem with the existing routing protocols, it is their lack of applicability to monitoring regional environments. There are many types of sensors in high-speed rail systems. Various sensors vary widely in terms of latency, transfer rate, data volume, etc. Inspired by the above requirements and the existing work, this paper proposes a heterogeneous network data aggregation model and adaptive routing algorithm for high-speed railway monitoring network based on reinforcement learning. The overall framework of the backbone routing communication protocol for high-speed railway state monitoring wireless sensor network is depicted in Figure 1, consisting of four components:Communication structure and data feature analysis, encompassing the analysis of node energy information, monitoring device characteristics, monitoring functional requirements, and data features;Analysis of monitoring targets and requirements, including aspects such as operational environment, track service status, and OCS (Overhead Catenary System) system status;Aggregation strategy and intelligent routing modeling, constructing an adaptive multi-objective optimization model based on the lifecycle and transmission delay requirements of different network objects;Adaptive routing algorithm, developing separate lifecycle and real-time evaluation models for monitoring tasks, dynamically adjusting multi-hop transmission paths to ensure the network fulfills both lifecycle and real-time requirements for various tasks simultaneously.

The main contributions of this paper can be summarized as follows:A lossless data aggregation transmission model for HSR networks is proposed, which can effectively reduce the amount of data in the network and reduce the energy consumption of data transmission;A Double-Q-value model based on data aggregation is proposed. Forn the two Q values, we consider the data aggregation degree, the remaining energy level, the link strength, the distance from the node to the sink, and the forwarding delay to consider the network lifetime and the real-time performance of data forwarding. The defined reward function can capture the dynamic changes in the network in real time and achieve dynamic control of the entire network with less overhead;An adaptive energy-saving routing algorithm based on Double-Q-values is proposed to classify HSR network devices according to their real-time requirements and life cycle requirements. An adaptive control algorithm is adopted for different business priorities. It meets the real-time requirements of the HSR network and prolongs the network’s life and improves service quality.

The work arrangement of this paper is as follows. Section 3 presents the typical structure of the high-speed railway monitoring network as well as the analysis of node requirements and introduces the data aggregation scheme. Section 4 describes in detail the proposed demand-aware energy-saving routing algorithm based on Q-learning. Section 5 discusses and analyzes the performance metrics of the proposed routing protocol through simulation experiments. The conclusions are summarized in Section 6.

## 3. System Profile and Overall Scheme

The system involved in this paper consists of the following four parts:Structure and characteristics of communication;Keeping track of objects and requirements;Policy of aggregation and intelligent routing design;Algorithm for adaptive routing.

### 3.1. Communication Structure and Characteristics

The high-speed train operating environmental monitoring system based on the wireless sensor network is simply called HSR-N, which is used for the completion, replenishment, or replacement of the transmission high-speed train operating environmental monitoring system. The proposed wireless sensor network can be roughly considered a linear network.

In HSR-N, its monitoring areas are wide and varied, and different monitoring targets have great differences. Therefore, relevant parameters and network technologies need to be designed for specific monitoring targets. With limited network energy, HSR-N needs an energy-efficient transmission protocol that meets its requirements.

### 3.2. Monitoring Objects and Requirements

The monitoring objects of the railway infrastructure monitoring system mainly include bridges, tunnels, roadbeds, contact networks, and rails. Obviously, different infrastructures have different monitoring challenges and requirements. For instance, in the health monitoring of bridges, the system needs to use high-precision sensors to monitor bridge structural cracks, deformations, and other conditions. Due to the non-maintainable nature of the internal structure of bridges, this requires bridge monitoring sensors to have a longer lifespan, with maintenance intervals preferably on a monthly basis. On the other hand, due to the slow changes in bridge structures, monitoring data does not require real-time updates. Therefore, bridge monitoring sensors have lower real-time requirements. Similar to bridges, rails, roadbeds, and tunnels have similar monitoring requirements, but their maintenance is relatively easier. Therefore, the lifespan requirements for sensors for the latter three are not as strict as those for bridges. Monitoring the contact network requires the use of temperature or strain sensors. Since the contact network is a vital component of the railway electrification system and directly affects train safety, sensors of this kind need to upload monitoring data promptly and have higher real-time requirements.

Based on practical project experience, this article categorizes the lifecycle requirements of monitoring equipment with maintenance cycles of one day, one week, and more than one month into three levels: low, medium, and high. Similarly, the real-time requirements of monitoring equipment with data sending frequencies higher than once every 10 min, between 10 min and 1 h, and more than 1 h are divided into three levels: high, medium, and low. Some monitoring objects and their corresponding characteristics are shown in Table 2. The real-time requirements and life cycle requirements of monitoring objects are different. We conduct a brief evaluation of their different functional requirements to provide support for the following work. The evaluation is mainly allocated based on the urgency of the data and the amount of data. For data types with high real-time requirements, we provide higher real-time evaluation to reduce delay; for periodic data with low real-time requirements, we base our calculations on the way the data are set up; the aggregation processing is designed to provide the network with a longer life cycle [26,27,28].

### 3.3. Aggregation Policy and Intelligent Routing Design

For most of the monitoring requirements of chronic changes, we only need to collect their data periodically and analyze it, while some monitoring requires only a large number of repeated measurements as the train passes by. For these data with low real-time requirements, we consider data aggregation on the transmission path to reduce the transmission burden of nodes to achieve the purpose of saving energy and prolonging the network life cycle. A lossless data aggregation model is adopted in this study. This means that the original data can be reconstructed by the sink node from the received aggregated data packets without any damage or loss of data. The data aggregation model is expressed as follows:(1)$$\begin{array}{c} \;\;\;\;\;DA\left\{Q_{i}^{t}\left(n\right)\right\}=\;\;\;\;\;\;\;\left\{\begin{array}{cc} U_{m}^{t}\times \log_{2}\left(DP_{i}\left(n\right)+1\right) & if_{}\;0<DP_{i}\left(n\right)\\ 0 & if_{}\;DP_{i}\left(n\right)=0. \end{array}\right. \end{array}$$

The data aggregation model is defined in this paper. The data buffer area of each sensor is partitioned, and each sensor maintains storage areas of multiple data types. For the same type of data, when the real-time requirement is low, the sensor performs data aggregation before forwarding to optimize the data volume. When a packet of data type t is transmitted in the network, each node transmits to the next hop after the interval $SI^{t}$. Obviously, the longer the waiting time for data aggregation, the greater the delay for the data packet to reach the sink node. It can be seen from Table 2 that different types of sensors have different real-time requirements for data. Therefore, in the design of routing protocols, the real-time requirements of different nodes need to be considered when energy-saving design is carried out [29,30]. This is explained in Section 4. The aggregation process design is shown in Figure 2 and model parameters are listed in Table 3.

If the sensor node $S_{i}$ is the next hop selected by $S_{j}$, $S_{k}$, and $S_{l}$, the t1-type aggregated data packets ${\mathrm{AD}}_{j}^{t1}\left(n\right)$ from the node $S_{j}$ and the t1-aggregated data packets ${\mathrm{AD}}_{k}^{t1}\left(n\right)$ of the $S_{k}$ node are stored together with the t1 data packets ${\mathrm{OD}}_{i}^{t1}\left(n\right)$ observed by the $S_{i}$ node from the surrounding environment. After aggregating into data packets ${\mathrm{AD}}_{i}^{t1}\left(n\right)$ in the t1 data type queue $Q_{i}^{t1}\left(n\right)$ of node $S_{i}$, they are sent to the next hop node $S_{p}$ of the t1 type. Similarly, the t2 type data packets ${\mathrm{AD}}_{k}^{t}2\left(n\right)$ from nodes $S_{k}$ and $S_{l}$, ${\mathrm{AD}}_{l}^{t2}\left(n\right)$ is in the t2 data of node $S_{i}$. The data packets ${\mathrm{AD}}_{i}^{t2}\left(n\right)$ are aggregated in the type queue $Q_{i}^{t2}\left(n\right)$ and sent to the next-hop node $S_{q}$ of type t2. After this aggregation process, the data of the same type is first aggregated and sent to the next-hop node with the same data type until the data packet is sent to the sink node to complete the aggregation and transmission of the data.

### 3.4. Adaptive Routing Algorithm

In the aggregation model described in Section 3.3, the aggregated data of each sensor node are sent to the optimal next-hop node, and the next-hop node selection is determined by the Q-learning adaptive algorithm proposed in this paper (see Section 4). The adaptive routing algorithm consists of three parts: (1) The sending node selects the node with the highest priority in the Q routing table to send the data packet; (2) The receiving node feeds back the reward value information to the sending node according to the received data packet; (3) The sending node accepts the reward value information and updates its Q routing table.

## 4. Model and Methodology of the Adaptive Protocol

Q-learning is a model-free reinforcement learning algorithm whose core is the Q-value and reward [31]. In order to maximize the network life cycle and meet the functional requirements of various types of sensors, this section presents an adaptive communication routing algorithm based on improved Q-learning, as shown in Figure 3.

The algorithm first defines dual Q-values using energy balance and real-time data transmission as metrics. It then introduces an adaptive weighting factor to optimize the design based on the requirements of the business. Next, it constructs reward models for energy saving and latency, and finally updates the Q-values based on the rewarded data [32]. The parameters and definitions used in this section are shown in Table 4.

### 4.1. Energy Model

Nodes adopt a periodic sleep/active work mode. The main energy consumption of nodes can be divided into two parts:Energy consumption in an active mode. We use $\omega_{a}$, which denotes the energy consumption rate in this mode.Energy consumption when nodes send and receive data.

We adopt the typical WSN energy consumption model to calculate the energy consumption of sending, receiving, and aggregated data, which is given by Equations (2)–(4), respectively.
(2)$$E_{T}\left(l\right)=\left\{\begin{array}{c} l\times E_{elec}+l\times \varepsilon_{fs}\times d^{2},d<d_{0},\\ l\times E_{elec}+l\times \varepsilon_{mp}\times d^{4},d\geq d_{0}, \end{array}\right.$$
(3)$$E_{R}\left(l\right)=l\times E_{elec},$$
(4)$$E_{D}\left(l\right)=l\times E_{ec},$$
where $E_{elec}$ represents the node’s energy consumption when sending, and $d_{0}$ represents the distance between nodes. When $d<d_{0}$, the node energy consumption is in the normal loss mode; when $d>d_{0}$, the node energy consumption is gradually reduced; $\varepsilon_{fs}$ and $\varepsilon_{amp}$ represent the energy at different distances; $E_{ec}$ is the energy expended for computation; *l* is the length of the data frame.

### 4.2. Double-Q-Value Learning Model

The Q-learning algorithm is a value function-based algorithm in RL, and for any finite Markov decision process, Q-learning can find an optimal policy. Q-learning involves an agent, a set of states *S*, and a set of actions *A*. By performing actions in the environment that cause the agent to move from one state to another, the action in a particular state is rewarded. That is, Q(s,a) is the expected reward for performing action *a*(ainA) in state *s*(sinS) at a given time [33]. The algorithm used in this study is shown in Figure 4.

### 4.3. Figures, Tables and Schemes

#### 4.3.1. State and Action

In the proposed double-Q-value learning model for high-speed railway wireless monitoring network routing, adjacent nodes exchange routing information in a cooperative way to ensure that nodes in the network can dynamically follow network changes and reduce the burden of maintaining the global routing table [34]. We define a sensor node’s node set *S*, action ainA, and action state set *A* as follows when it sends a specific type of data to the next hop node:(5)$$\begin{array}{c} S=\left\{s_{1},s_{2},\cdot \cdot \cdot ,s_{n}\right\},\\ A=\left\{A_{1},A_{2},\cdot \cdot \cdot ,A_{n}\right\},A_{i}=\left\{a_{j}=s_{j}|s_{j}\in F_{si}\right\}, \end{array}$$
where *n* is node number and $F_{si}$ is the set of forwarding nodes of node $s_{i}$.

#### 4.3.2. Initialization of the Double-Q-Values

In Q-learning, the forwarding of data between nodes uses a Q-table to find the best action, where the Q-value is the expectation of nodes when forwarding [35]. In the double Q-value model designed in this paper, the action value functions are divided into life cycle functions and real-time functions $Q_{L}$ and $Q_{T}$. $Q_{L}$ consists of three parts: data aggregation degree, node energy status, and link strength. The first part aims to increase the aggregability of forwarded data packets and reduce the data size to reduce the energy loss caused by data transmission. The second part avoids selecting energy. Nodes with low values are forwarded, and the third part aims to reduce communication overhead and save energy. $Q_{T}$ consists of two parts: the number of hops reaching the sink node and the forwarding delay estimation, both of which aim to ensure the real-time performance of the data packet reaching the sink node. As shown in (6), double-Q-values are initialized as a weighted sum of the probabilities of their respective parts.
(6)$$\begin{array}{c} Q_{L}\left(s,a\right)=DA\left(s,a\right)+E_{s}^{a}+L_{s}^{a},\\ Q_{T}\left(s,a\right)=d\left(a,\sin k\right)+T_{s}^{a}, \end{array}$$
where DA(s,a) denotes the degree to which node s aggregates data to the node indicated by its action a; $E_{s}^{a}$ is the remaining energy of the node pointed to by action a. The link strength between node s and the node pointed to by action a is represented by $L_{s}a$. d(a,sink) is the distance between the node indicated by action a and the sink. $T_{s}^{a}$ is forwarding time from node s to the node pointed to by its action a. Before starting, the Q-values are initialized only by the initial energy and the distance to the sink, and other parameters are updated after running.

#### 4.3.3. Double Q-Value Update

In this paper, $Q_{L,T}\left(s,a\right)$ defines the possibility of state *s* acting *a* and provides various types of businesses with a Q-table based on their business requirements, which is defined as follows:(7)$$Q_{L,T}\left(s,a\right)=\left[\begin{array}{c} Q_{L,T}^{t1}\left(s,a\right)\\ Q_{L,T}^{t2}\left(s,a\right)\\ \vdots \\ Q_{L,T}^{tn}\left(s,a\right) \end{array}\right].$$
Among them, $t_{1}-t_{n}$ is the business type, $Q_{L}$ is the life cycle measurement of an action, and $Q_{T}$ is the real-time measurement of an action.

When a node selects the optimal next hop in its Q-table to send a packet, it obtains a reward from the receiving node and updates its Q-value accordingly. The new Q-value is (8),
(8)$$Q_{L,T}\left(s,a\right)=Q\left(s,a\right)+\alpha \left\{R_{L,T}-\gamma \cdot Q\left(s,a\right)\right\},$$
where α is the learning rate and γ is the discount factor for the future reward.

#### 4.3.4. Explore Strategies

Usually, action selection relies only on the highest Q value, but this fixed selection can become stuck in a local optimum. To achieve this, we use an epsilon-greedy algorithm that makes it possible to escape local optima with partial probability.
(9)$$a_{*}|s=\left\{\begin{array}{cc} \arg\max Q(s,a) & with\;probablity\;1-\epsilon \\ any\;action\;a & with\;probablity\;\epsilon \end{array}\right..$$

#### 4.3.5. Future Rewards

In this stage, rewards are given for the action performed in the previous step, which can be divided into three situations.

(1) The node receiving data packets is not the sink node, and the energy level is normal. We assign each component of double-Q-values its own reward scheme, calculated as follows:(10)$$DA_{nor}=\left\{\begin{array}{cc} \frac{Q_{s^{\prime }}^{t}\left(n\right)}{AD_{s^{\prime }}^{t}\left(n\right)}-1 & if\frac{Q_{s^{\prime }}^{t}\left(n\right)}{AD_{s^{\prime }}^{t}\left(n\right)}-1<r_{DA}^{\max}\\ r_{DA}^{\max} & else\frac{Q_{s^{\prime }}^{t}\left(n\right)}{AD_{s^{\prime }}^{t}\left(n\right)}-1>r_{DA}^{\max} \end{array}\right.,$$
(11)$$E_{nor}=E_{r}^{s^{\prime }}/E_{i}^{s^{\prime }},$$
(12)Ls(si,sj)=RecsipRecsjob,Recsjob=pr(d)pr(d0)db=−10ρlog(dd0)+Xdb,Lnor=ls(s,s′),
(13)$$\begin{array}{c} radv=d\left(s,si\;nk\right)-d\left(s^{\prime },si\;nk\right),\\ radv_{avg}=\sum_{i=1}^{n}\left|radv_{i}\right|/n,\\ A_{nor}=radv/radv_{avg}. \end{array}$$
(14)$$T_{nor}=T\left(s,s^{\prime }\right)/T_{avg}$$

(2) The receiving node is sink. The reward is a constant $R_{s}$ when the chosen action sends the data packet to the sink node.
(15)$$R=R_{s}.$$

(3) The receiving node is not sink, and the energy level is lower than the average energy of network nodes. In order to maintain the performance and life of the network, we suggest not assigning the node the function of forwarding data when the energy level is too low to ensure that the basic monitoring service of the network is normal [36,37]. Therefore, we offer a negative reward to avoid packets from the neighbor nodes.
(16)$$R=-R_{e}.$$

Based on the above definition, corresponding to the double-Q-values, the future reward should also be divided into two parts to offer different rewards for its life cycle and real-time performance. At the same time, in order to avoid increasing the probability of forwarding to nodes far away from the sink, it is necessary to provide a discount value to the reward. Rewards $R_{L}$ and $R_{T}$ for state s are calculated as follows:(17)$$R_{L}=\left\{\begin{array}{cc} \alpha_{l}\times R_{DA}+\beta_{l}\times R_{E}+\gamma_{l}\times R_{L} & s^{\prime }\neq si\;nk\\ R_{s} & s^{\prime }=si\;nk\\ -R_{e} & E_{s^{\prime }}\;is\;low \end{array}\right.,$$
(18)$$R_{T}=\left\{\begin{array}{cc} \alpha_{t}\times R_{A}+\beta_{t}\times R_{T} & s^{\prime }\neq si\;nk\\ R_{s} & s^{\prime }=si\;nk\\ -R_{e} & E_{s^{\prime }}\;is\;low \end{array}\right..$$

### 4.4. Adaptive Routing Protocol Based on Double-Q-Values

In order to meet different business needs, this section proposes an adaptive weighting scheme based on the proposed double-Q-value model and the principles of maximizing network lifetime and adaptively meeting the functional requirements of various types of sensors, trying to consider both energy saving and delay reduction. Two methods are used to adapt to different business objectives. The multi-objective function formula is as follows:(19)$$\max Q=\max(\alpha_{w}\times Q_{L}+\left(1-\alpha_{w}\right)\times Q_{T}),$$
where $\alpha_{w}$ is a business adaptive weighting factor used to adjust the weight of a life cycle and real-time goals. As shown in Table 2, different services in the system have different requirements for life cycle and real-time performance. When the real-time performance requirement of the monitoring object is low, that is, the frequency of the device sending data is less than once an hour, it means that the object has higher requirements for the continuity of data transmission. The adaptive weight factor increases with the improvement of life requirements; and when the real-time requirement of the monitored object is high, that is, the frequency of the device sending data is higher than once every 10 min, the algorithm assigns priority to meeting its real-time requirements. Therefore, the design of the adaptive weight factor is as follows: in this formula, low, medium, and high values are 0.2, 0.5, and 0.8.
(20)αw=log(η×(DlsDts))log(η×max(DlsDts))Dts=20.5×log(η×(DlsDts))log maxDlsDts+0.5×11+e(Dls−Dts)×Dts(ξ−(Dls−Dts)×Dts)Dts=0.511+e(Dls−Dts)×Dts(ξ−(Dls−Dts)×Dts)Dts=0.8,
where the life cycle demand amplification factor η and the real-time demand amplification factor ξ are defined as follows:(21)$$\begin{array}{cc} \eta =\frac{\max D_{l}^{s}}{\max D_{t}^{s}} & 0<D_{l}^{s}<1,0<D_{t}^{s}<0.5, \end{array}$$
(22)$$\begin{array}{cc} \xi =\max(D_{t}^{s}-D_{l}^{s}) & 0<D_{l}^{s}<1,0.5<D_{t}^{s}<1. \end{array}$$

## 5. Performance Comparison and Validation

In this section, we compare and analyze the performance of the proposed Double Q-value Adaptive Aggregation Routing Protocol (DQAAR) in terms of energy consumption, network lifetime, transmission delay, and data retransmission energy loss. At present, there is little research on multi-objective optimization of high-speed railway wireless sensor network delay and life cycle. Due to MATLAB’s powerful computational capabilities and rich reinforcement learning toolbox, the improved Q-learning algorithm proposed in this paper can be easily implemented. Therefore, this paper uses MATLAB R2018b to realize the simulation environment and compares it with the other three excellent routing protocols [38]. They are MTECR, AUMRP, and MVECR.

### 5.1. Parameters Configuration

In this study, the node communication energy consumption adopts the space energy loss model, and the simulation parameter configuration is shown in Table 5.

The initial energy of the sensor nodes involved in this paper is 0.5j, and the energy of sink nodes is unlimited. According to the node types and business requirements in Table 1, the network model is constructed proportionally to verify the performance of the adaptive routing model in this paper. In the comparative analysis before, we introduced a few concepts about performance indicators. (1) FND (the time at which the first node dies); (2) HND (time when half of the nodes die); (3) CP Index (Comprehensive Performance Index); the utility of a high-speed railway monitoring network is determined by its life cycle and real-time performance. In this paper, we build a complete evaluation model for life cycle and real-time performance:(23)$$U=\lambda \times \frac{L}{\max(L)}+\left(1-\lambda \right)\frac{\min(T)}{T}.$$

In the table, *U* is the comprehensive performance index, *L* is the life cycle, and *T* is the delay.

### 5.2. Results and Discussion

In order to verify the effectiveness of the adaptive routing protocol proposed in this paper, we first verify the effect of extending the network life cycle of each protocol, and then simulate environmental changes and different application scenarios by changing some system parameters. Several aspects, such as extending the comprehensive efficiency index, are compared with the three routing protocols mentioned above, which verifies the superiority of DQAAR.

#### 5.2.1. Lifetime Evaluation

Figure 5 shows the change in the number of surviving nodes for each routing protocol with the network running time. For the high-speed railway monitoring network, the main task is to collect as much on-site information as possible to ensure the safety of railway operation, so we first focus on the life cycle of the network. Figure 5a shows that DQAAR has a longer running time than MTECR, MVECR, and AUMRP under the same conditions. Because MTECR only pays attention to the overall energy loss of the network and does not care about the energy balance of the network, ithe death time of each node occurs in about 1000 s. MTECR, MVECR, and AUMRP impose some constraints on the overall energy consumption balance of the network, but while prolonging the death time of the first node, they also cause a large area of low-energy nodes to die in the network around 1500 s.The DQAAR proposed in this paper delays the death time of the first node in the network to about 2000 s, and then there is no continuous death of large-scale nodes, but a relatively slow trend is maintained. This is because DQAAR offers dynamic rewards based on data aggregation while paying attention to the balanced use of network energy, and each node learns the best next-hop node. Efficient data aggregation paths greatly reduce the amount of data transmitted over the network, which greatly delays node death times. Figure 5b shows the FND and HND data of the four routing protocols, respectively. The FND and HND of the worst-performing MTECR are 985 and 2024, respectively, and the AUMRP data of FND are only 1565 and 2155, although it is effective. The node death time is delayed, but the process from the death of the first node to the death of half of the nodes is not very slow. The FND and HND of the DQAAR proposed in this paper are 1982 and 3016, respectively. It can be seen that DQAAR provides better answers in terms of balancing network energy consumption and improving network life.

#### 5.2.2. Latency Time

Figure 6 shows the real-time performance of DQAAR and the three protocols mentioned above. MTECR achieves the best real-time performance with an average latency of 1780 ms, but this is due to its advantage of reducing hop count at the expense of lifetime. In the initial stage of the network, DQAAR is in the parameter adjustment stage with significant delays. After a period of learning, its real-time performance is greatly improved, which is not much different from MTECR. AUMPR and MVECR excessively pursue the balance of energy consumption between nodes and achieve node energy balance by increasing the number of link hops. Long link hops greatly increase system latency.

Compared with the other three routing protocols, the DQAAR delay in the stable operation stage has better stability performance. That is to say, the initial high delay is caused by the exploration behavior of DQAAR. In the stable operation stage, the real-time performance of DQAAR far exceeds that of MVECR and AUMRP, achieving good real-time performance.

#### 5.2.3. Scenario Analysis

Case 1 verifies the impact of network size changes on its performance and the performance of each routing protocol. Among them, a single data packet on the network data = 200 bits. Figure 7 shows the performance of the life cycle, delay, and overall energy efficiency of each routing protocol when the distance from the head end to the end of the network changes from d = 100 m to d = 500 m. Figure 7a shows that with the increase in the network range, the energy balance of the network system is destroyed, which also causes the life cycle of each routing protocol to decrease significantly with the increase in d, but MTECR, AUMRP, MTECR, MVECR and DQAAR have better performance in extending the network life cycle. This is because MTECR minimizes the overall energy consumption of the network as an optimization goal, and the number of link hops is significantly reduced compared to the other three algorithms. This may lead to the emergence of local hotspots in the network, which may affect the entire network lifecycle. When the network range is expanded to 500 m, its lifetime is still about 30% higher than the worst performing MTECR. Figure 7b shows the delay performance of each routing protocol as the network range increases, and the time for each routing protocol data packet to reach the sink node gradually increases as the communication distance increases. Among them, MVECR and AUMRP lack the constraints on the delay, which leads to the rapid increase in the delay when the distance increases, and DQAAR reduces the delay through dynamic learning so that the network can obtain good real-time performance. In order to obtain better real-time performance, MTECR reduces the number of hops of data packet forwarding in the network, which greatly increases the energy consumption and shortens the network life.

Obviously, the DQAAR proposed in this paper is ahead of AUMRP, MTECR, and MVECR in energy efficiency. When the network range is small, the life cycle of AUMRP and MVECR is close to that of DQAAR, and their comprehensive energy efficiency is also close to that of DQAAR. The life cycle index and delay index of the network system using each protocol are shown in Figure 8 and Figure 9.

Case 2 verifies the impact of changes in the amount of data in the network on its performance and the performance of the four routing protocols. Among them, the network range is d = 200 m. Figure 10 shows the network life cycle, delay and comprehensive energy efficiency level of the DQAAR proposed in this paper and the other three routing protocols (AUMRP, MVECR, and MTECR) when the single data packet size of the network node changes from Data = 100 bit to Data = 600 bit. As shown in Figure 10a, an increase in the amount of data in the network is accompanied by a rapid decrease in its life cycle, because the sending and receiving of data consumes the most energy in sensor nodes. MVECR pays too much attention to the energy consumption balance of each node in the network, but it increases its total energy consumption, and the life cycle has a disadvantage compared with AUMRP. The DQAAR transmission path planning based on data aggregation effectively reduces the amount of data in the network and thus prolongs the life cycle of the network, and its performance is higher than the other three routing protocols. The delay of the network system using DQAAR is smaller than that using AUMRP and MVECR but slightly larger than that using MTECR, and the delay increases with the increase in data volume, which is caused by the increase in transmission time caused by the increase in data volume. Data latency increases. Also, as the amount of data increases, the trend of delay growth for DQAAR is not as fast as it is for AUMRP and MVECR. This shows that the data transmission delay is reduced enough by the data aggregation strategy used in this paper to make up for the time it takes to aggregate the data. It can be seen in Figure 10c that DQAAR can effectively reduce the amount of data when the amount of data in the network increases, while the delay does not cause a significant change. Its comprehensive energy efficiency is much greater than that of the other three routing protocols. The cycle and delay maintain stable performance with the increase in data volume.The lifetime index and delay index of the network system using each protocol are shown in Figure 11 and Figure 12.

Finally, the simulation verification results can be summarized into the following three points:

(1) Compared with AUMRP, the life of the network system using DQAAR is improved to a certain extent, and both MVECR and MTECR are improved to a certain extent, which effectively prolongs the dead time of the first node in the network system. It ensures the balance of network energy consumption and allows the longer survival of nodes with heavy loads in the network when the energy level is low to ensure the monitoring quality of the network.

(2) In the high-speed railway monitoring system, the life cycle of the network and the delay of data transmission are both important performance indicators, and the single-objective network optimization algorithm is difficult to meet the actual needs. The adaptive routing algorithm based on double-Q-values proposed in this paper can effectively improve the network life and obtain good real-time performance. The comprehensive energy efficiency index is used to evaluate the routing protocol and verify the superiority of DQAAR in these two aspects.

(3) The design of the adaptive operator and Q-value in this paper comes from the business requirements of each sensor in the high-speed railway monitoring network system. In different application scenarios, the adaptive operator and Q-value can be designed differently. This ensures the multi-scene adaptability of the adaptive model based on double Q-values established in this paper.

## 6. Conclusions

In this paper, we propose a Double-Q-value-based adaptive routing algorithm (DQAAR) for the business requirements of high-speed railway monitoring network systems. The proposed method is different from most of the existing methods and offers contributions described below.

First, we propose a Double-Q-value model based on data aggregation. For the two Q-values, we consider the data aggregation degree, the remaining energy level, the link strength, the distance from the node to the sink, and the forwarding delay to consider the network lifetime and the real-time performance of data forwarding. The defined reward function can track the network’s changes in real time and keep the whole thing under control with less work.

Second, an adaptive weight is proposed based on the different requirements of each service for network lifetime and real-time performance. This makes the algorithm proposed in this paper better able to adapt to different situations.

Finally, the algorithm proposed in this paper is verified in different scenarios. The results show that DQAAR is better than AUMRP and MVECR in achieving network energy balance and prolonging network life, and its real-time performance is also better than that of these two routing protocols. Compared with MTECR, although the routing protocol proposed in this paper is slightly insufficient in real-time performance, it is far better than MTECR in extending network life. From the point of view of overall energy efficiency, the DQAAR that is proposed in this paper is a lot better than other routing protocols.

## Figures and Tables

**Figure 1 sensors-23-07393-f001:**
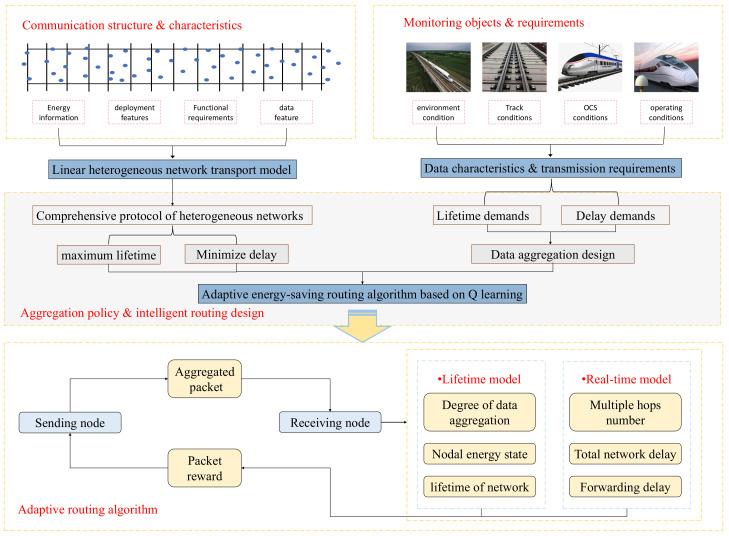
Overall scheme of adaptive routing protocol for high-speed railway monitoring network based on reinforcement learning.

**Figure 2 sensors-23-07393-f002:**
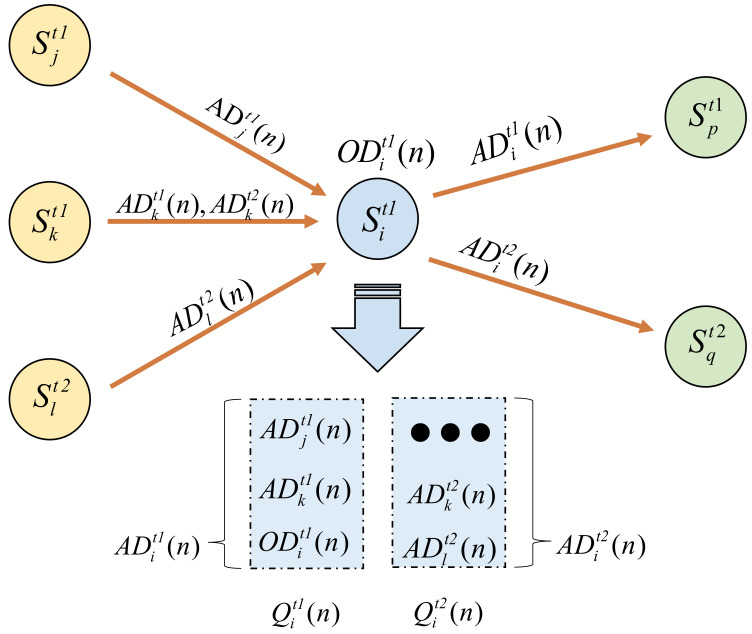
Data aggregation transmission model.

**Figure 3 sensors-23-07393-f003:**
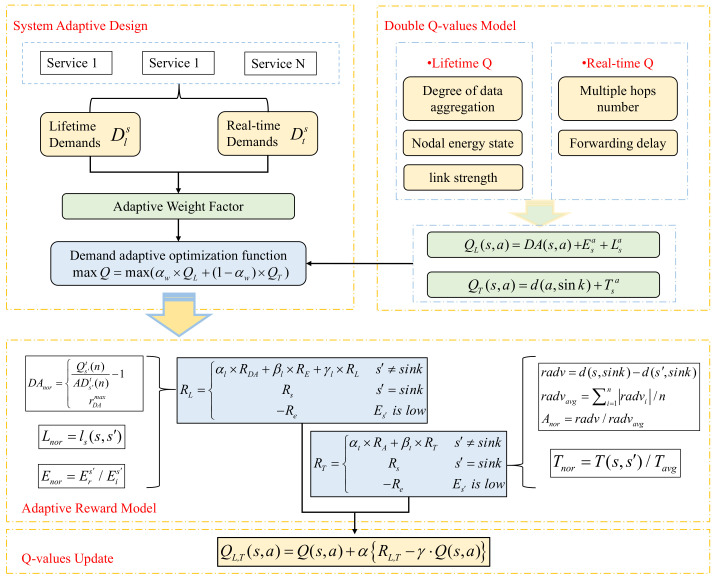
Double-Q-value adaptive algorithm framework.

**Figure 4 sensors-23-07393-f004:**
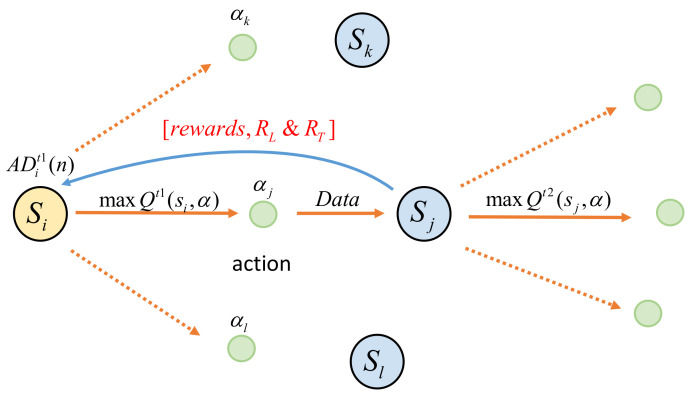
Application of double-Q-learning algorithm in transmission between nodes.

**Figure 5 sensors-23-07393-f005:**
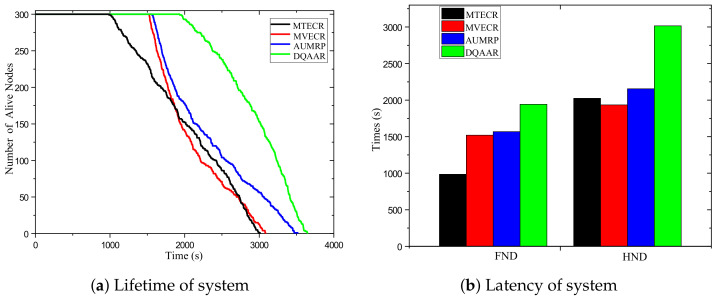
Protocol life cycle, FND and HND performance.

**Figure 6 sensors-23-07393-f006:**
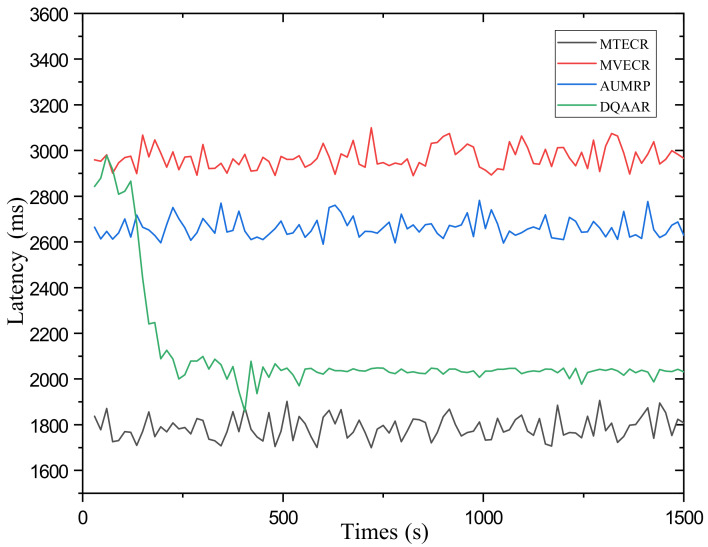
Delay performance of each protocol.

**Figure 7 sensors-23-07393-f007:**
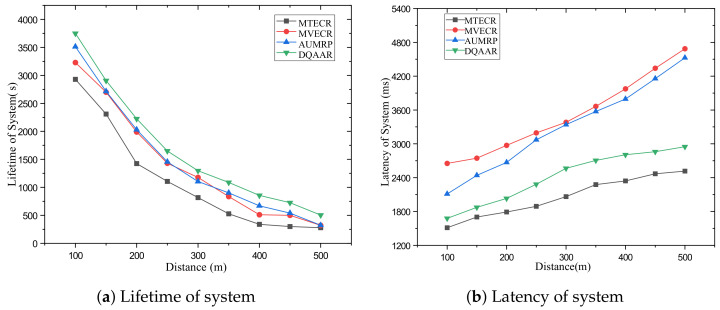
Performance of each protocol in Scenario 1.

**Figure 8 sensors-23-07393-f008:**
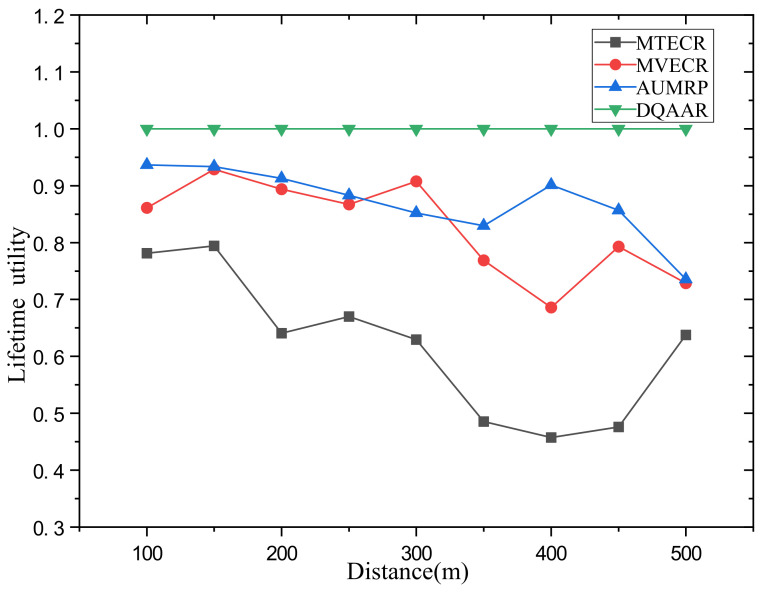
System lifetime utility of each protocol in Scenario 1.

**Figure 9 sensors-23-07393-f009:**
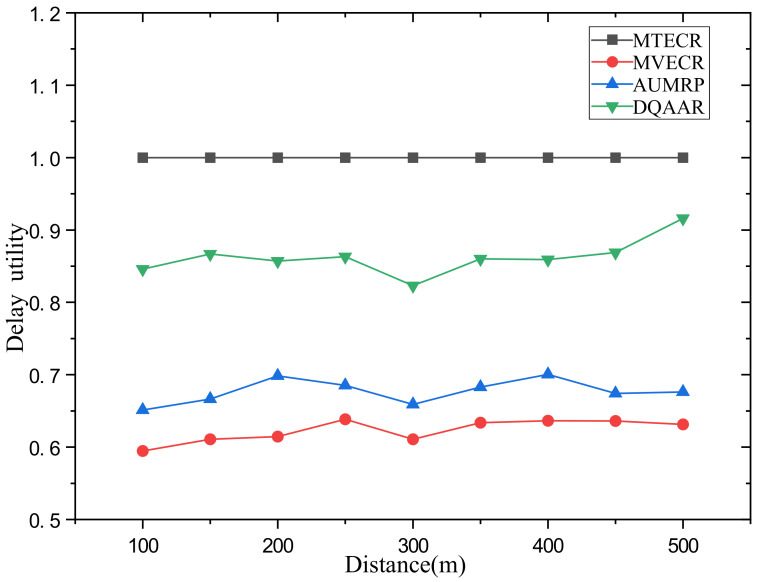
System delay utility of each protocol in Scenario 1.

**Figure 10 sensors-23-07393-f010:**
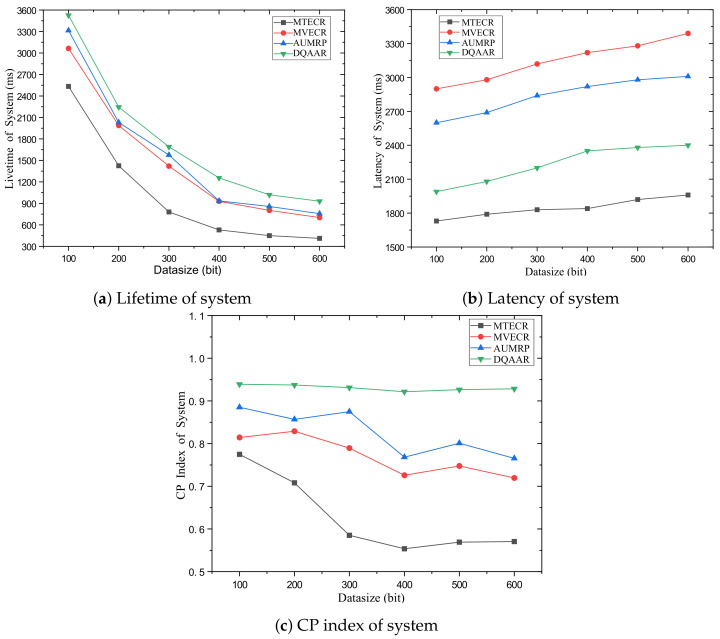
Performance of each protocol in Scenario 2.

**Figure 11 sensors-23-07393-f011:**
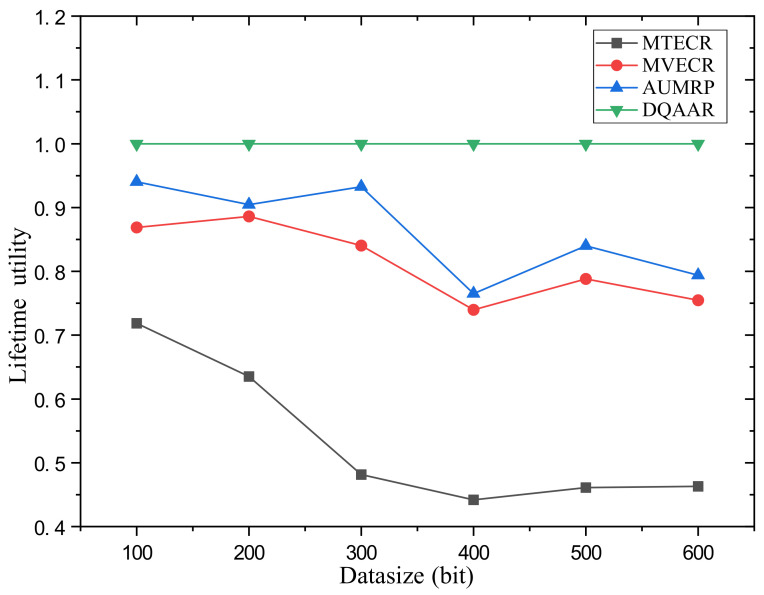
System lifetime utility of each protocol in Scenario 2.

**Figure 12 sensors-23-07393-f012:**
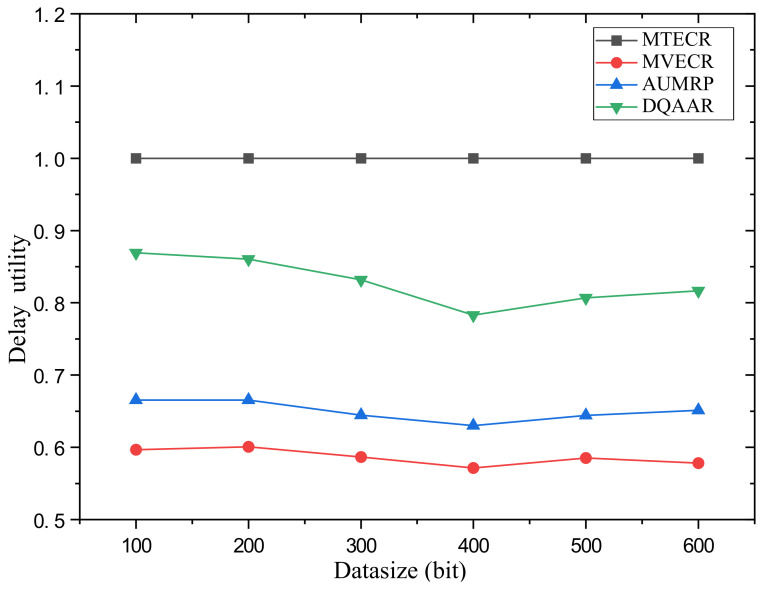
System delay utility of each protocol in Scenario 2.

**Table 1 sensors-23-07393-t001:** Existing methods and their characteristics.

Projects	Description	Contribution	Structure
Hybrid Energy Efficient Distributed Cluster (HEEDR) [18]	The cluster head is selected based on the remaining energy of the node and the cost of its communication	The residual energy-based strategy is used in both intra-cluster communication and inter-cluster head communication, improving network lifetime	Mesh
Energy saving distributed scheduling algorithm (CLU-DDAS) [19]	An energy-efficient distributed scheduling algorithm based on a novel cluster aggregation tree is proposed to minimize delay	Reduced data transfer delays in the network while singing network longevity	Tree
Routing Protocol to Minimize Total Network Energy Consumption (MTECR) [20]	Reducing the energy consumption in the network by reducing the number of data forwarding hops	Transmission of data with the minimum number of hops reduces forwarding energy consumption and transmission delay	Linear
Minimize variance of network energy consumption (MVECR) [21]	Minimize the energy consumption variance of each node in the network to achieve the purpose of improving the network lifetime	Minimizing the variance of energy consumption balances the energy consumption of each node in the network and increases the network lifetime	Linear
Distributed Energy Efficient Cluster Routing Protocol (DEECR) [22]	A routing protocol for heterogeneous networks is proposed, which selects cluster heads based on the ratio of remaining energy to the average energy level of the network	The strategy of selecting cluster heads by residual energy and average energy level successfully improves the energy efficiency of the network	Mesh
Adaptive Optimization of Multi-Hop Communication Protocol (AUMRP) [23]	Control the transmission power of each node and minimize the energy consumption of the node to achieve the purpose of improving the life of the network	Constrains the maximum energy consumption of nodes in the network so that the energy of nodes is preserved and the life of the network is improved	Linear
Energy Efficient Unequal Clustering Routing Protocol (EEUCR) [24]	Inhomogeneous clustering and dual-cluster head techniques are used to solve hotspot problems, and a hybrid rotation strategy based on node time and energy is also proposed to reduce energy consumption	Mitigates hotspot issues in the network with Rotational Forwarding	Mesh
Improved energy Efficient Cluster Head Selection Routing Protocol (IEECHS-WSN) [25]	Elect two cluster heads within a cluster and have them responsible, respectively, for data transmission and data fusion tasks, thereby extending the network lifespan	The strategy of selecting dual cluster heads and conducting data fusion reduces the amount of network data transmission and reduces network energy consumption	Mesh

**Table 2 sensors-23-07393-t002:** Monitoring objects and characteristics.

Monitoring Objects	Sensor Type	Life Cycle Demands	Real-Time Demands
Longitudinal stress of steel rail	Ultrasonic sensor	high	low
Rail deformation	Deformation sensor	high	low
Rail integrity	Ultrasonic sensor	medium	medium
Rail wear	Video monitoring	low	low
Track switch extension pitch adjuster	Fiber grating strain sensor	high	high
Rail stiffness	Rail inspection car	low	low
Foreign body contamination limit	Video monitoring	high	high
Track foundation submerges	Leica total station monitoring system	medium	low
Track slope condition	Laser laser scanner and fiber grating strain sensor	medium	medium
Lead power supply system	Infrared temperature sensor, fire detector, temperature and humidity sensor	high	high
Bow net service state	Acceleration sensor, ratchet deviation Angle sensor, cable clip temperature sensor	high	high
Suspension tension, elasticity and vibration	Tension measurement sensor, wire vibration sensor, elastic measurement sensor	medium	medium
Pantograph image recognition	Image and Video Signal Processing	high	high
Geological disasters	Seismic detector, landslide detection	low	low
Meteorological disaster	Laser monitoring equipment, video and image processing	low	high
Meteorological watch	Temperature sensor and humidity sensor	high	high

**Table 3 sensors-23-07393-t003:** Aggregation model parameters.

Parameters	Symbol
Sensor node i with sensor type t	$S_{i}^{t}$
Sensing interval for type t	$SI^{t}$
Aggregation data by sensor node i for type t at time step n	${\mathrm{AD}}_{i}^{t}\left(n\right)$
Observed data by sensor node i for type t at time step n	${\mathrm{OD}}_{i}^{t}\left(n\right)$
Queue state of sensor node i for type t at time step n	$Q_{i}^{t}\left(n\right)$
Unit packet size of type t for aggregation model m	$U_{m}^{t}$
Number of packets in the aggregation queue of node i at step n	$DP_{i}\left(n\right)$

**Table 4 sensors-23-07393-t004:** Protocol parameters and definitions.

Parameter	Representation
$E_{elec}$	Energy consumed to transmit a unit of bit data
$\varepsilon_{fs}$	Power amplifier normal loss
$\varepsilon_{amp}$	Power amplifier for multipath attenuation
$d_{0}$	Distance constant
$E_{ec}$	The energy spent for computation
DA(s,a)	The degree of data aggregation of node s to the node pointed to by its action a
$E_{s}^{a}$	The remaining energy of the node pointed to by action a
$L_{s}^{a}$	Link strength between node s and the node pointed to by action a
d(a,sink)	The distance from the node pointed to by action a to the sink
$T_{s}^{a}$	Forwarding time from node s to the node pointed to by its action a
$DA_{nor}$	Data Aggregation normalized value
$E_{nor}$	Residual energy normalized value
$A_{nor}$	The normalized value of the number of hops to the sink node
$T_{nor}$	Forwarding delay normalized value
$L_{nor}$	Link strength normalized value
Recsp	Received signal power
$\alpha_{w}$	Adaptive weight factor

**Table 5 sensors-23-07393-t005:** Aggregation model parameters.

Parameter	Representation	Value
$E_{elec}$	Energy consumed to transmit a unit of bit data	50 nJ/bit
$\varepsilon_{fs}$	Power amplifier normal loss	10 pJ/bit/m${}^{2}$
$\varepsilon_{amp}$	Power amplifier for multipath attenuation	0.0013 pj/bit/m${}^{4}$
$E_{in}$	Initial energy of nodes	0.5 j
$d_{0}$	Distance threshold	87 m
$D_{l}$	Lifetime requirements	[0.2,0.5,0.8]
$D_{t}$	Real-time requirements	[0.2,0.5,0.8]
$r_{DA}^{\max}$	Maximum aggregation reward	1
α,γ	Learning rate and discount factor	0.8,0.9
η	Magnification factor of lifetime	9
ξ	Magnification factor of real time	0.8
*R*	Network range	200 m
Data	Network single packet size	200 bit

## Data Availability

The data presented in this study are available in the article.

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
