# Peer review of "Energy-Saving Adaptive Routing for High-Speed Railway Monitoring Network Based on Improved Q Learning"

_sensors, 2023, doi:10.3390/s23177393_

Round 1

Reviewer 1 Report

The paper deals with two important issues (meeting the network life time and the delay constraint)  in high speed railway wireless sensor networks. To optimize the two conflicting objectives, it proposes a double Q value based model (DQAAR) on data aggregation and routing. It evaluates the proposed scheme by comparing it with the exiting algorithms. According to the MATLAB based evaluation results, the proposed one outperforms the existing ones. The paper is well organized and presented, but it has the following issues to be addressed:

1.     It would be better to justify why MATLAB is used for the evaluation even if there are number of network simulators

2.     (line number 310). “the energy level is too low”. It’s ambiguous and more objective (and possibly quantitative) expression is preferred.

3.     Even though the paper presents the comparison of METCR and DQAAR, it’s not convinced that the users prefer DQAAR to METCT.

4.     It’s not clear that the reference section includes the references for the existing algorithms while Table A.1 summarizes them.  

5.     “objective” is an appropriate expression in line 328- 334

6.     Minor comments(in part):

A.      Case sensitive issues

                            i.          “we Based on how” in line number 168

                           ii.          “node I at step n” in Table 1

                         iii.          “A losseless” in line number 187

                         iv.          “data type T is” in line number 194

                           v.          “Eec is “ in line number 250

                         vi.          “the Set of” in line number 267

B.      etc

                            i.          “Serve.” In line number 54

                           ii.          “studies.Finally” in line 15

                         iii.          “proposed.From” in line 122

                         iv.          “Sec.3”, “Sec.4”, “Sec.5” in line 133-138 vs. “Section 3.3” in line 214

please refer to "Comments and Suggestions for Authors
"

Author Response

请参阅附件

Reviewer 2 Report

Why this paper is available online already at 3-4 different locations like 

1. https://europepmc.org/article/ppr/ppr690946

2. https://www.researchgate.net/publication/372346532_Energy-Saving_Adaptive_Routing_for_High-Speed_Railway_Monitoring_Network_Based_on_Improved_Q_Learning

3. https://www.preprints.org/manuscript/202307.0947/v1

State the novelty and inventiveness of the paper in intro section when compared with "An intelligent and autonomous sight distance evaluation framework for sustainable transportation" and "Energy-Saving Optimization Method of Urban Rail Transit Based on Improved Differential Evolution Algorithm"

Analytical section and Result section is good in the paper but reason for better performance seems missing from the document.

Explanation of figures is missing. Authors should add it, like Figure 1 needs a detailed explanation of working. Similarly the algorithm portion is much more desirable here in the manuscript.

Needs to edit the manuscript at some places for correcting grammatical errors.

Reviewer 3 Report

Is the designed monitoring network special for all infrastructures or vehicles in railway systems? The object should be more specifically revealed.

Some statistics should be given to demonstrate the reliability of the developed monitoring network in the abstract.

Some sentences are strange. Like ‘Wireless sensor networks’ biggest problem is saving energy ’ Is saving energy not a pro but a problem?

In the introduction, it deserves to mention the variety of the infrastructures like the track-track [1] and OCS [2], as well as the complex working condition of railway systems, like the wind load [3] and earthquakes [4]. The faults are the direct consequence of these factors.

[1] Zhai, W. (2020). Vehicle–Track Coupled Dynamics. In-Vehicle–Track Coupled Dynamics. Springer Singapore. https://doi.org/10.1007/978-981-32-9283-3

[2] Song, Y., et al. (2021). A spatial coupling model to study the dynamic performance of pantograph-catenary with vehicle-track excitation. Mechanical Systems and Signal Processing, 151, 107336. https://doi.org/10.1016/j.ymssp.2020.107336

[3] Fuchuan Duan, et al (2023). Study on Aerodynamic Instability and Galloping Response of Rail Overhead Contact Line Based on Wind Tunnel Tests. IEEE Transactions on Vehicular Technology. https://doi.org/10.1109/TVT.2023.3243024

[4] Miyamoto Takefumi, et al "Running safety of railway vehicle as earthquake occurs." Railway Technical Research Institute, Quarterly Reports 38.3 (1997).

The term ‘OCS’ is not explained.

Different infrastructures may have different challenges to be monitored in Fig. 1. Please specify them and

Some format issues should be sorted out. For instance, the first letters of some section titles are not in uppercase.

Round 2

Reviewer 1 Report

The authors have addressed all the issues in the previous manuscript.

Reviewer 3 Report

No other issues. This paper is recommended for publication.